# Transcriptome-wide association study and eQTL colocalization identify potentially causal genes responsible for human bone mineral density GWAS associations

Basel Maher Al-Barghouthi[1,2], Will T Rosenow[1], Kang-Ping Du[3], Jinho Heo[4], Robert Maynard[5], Larry Mesner[1,6], Gina Calabrese[1], Aaron Nakasone[7], Bhavya Senwar[8], Louis Gerstenfeld[9], James Larner[3], Virginia Ferguson[8], Cheryl Ackert-Bicknell[5], Elise Morgan[7], David Brautigan[4], Charles R Farber[1,2,6]*

[1]Center for Public Health Genomics, School of Medicine, University of Virginia, Charlottesville, United States; [2]Department of Biochemistry and Molecular Genetics, School of Medicine, University of Virginia, Charlottesville, United States; [3]Department of Radiation Oncology, University of Virginia, Charlottesville, United States; [4]Department of Microbiology, Immunology, and Cancer Biology, School of Medicine, University of Virginia, Charlottesville, United States; [5]Department of Orthopedics, Anschutz Medical Campus, University of Colorado, Aurora, United States; [6]Department of Public Health Sciences, School of Medicine, University of Virginia, Charlottesville, United States; [7]Department of Mechanical Engineering, Boston University, Boston, United States; [8]Department of Mechanical Engineering, University of Colorado Boulder, Boulder, United States; [9]Department of Orthopaedic Surgery, Boston University Medical Center, Boston, United States

*For correspondence: crf2s@virginia.edu

**Abstract** Genome-wide association studies (GWASs) for bone mineral density (BMD) in humans have identified over 1100 associations to date. However, identifying causal genes implicated by such studies has been challenging. Recent advances in the development of transcriptome reference datasets and computational approaches such as transcriptome-wide association studies (TWASs) and expression quantitative trait loci (eQTL) colocalization have proven to be informative in identifying putatively causal genes underlying GWAS associations. Here, we used TWAS/eQTL colocalization in conjunction with transcriptomic data from the Genotype-Tissue Expression (GTEx) project to identify potentially causal genes for the largest BMD GWAS performed to date. Using this approach, we identified 512 genes as significant using both TWAS and eQTL colocalization. This set of genes was enriched for regulators of BMD and members of bone relevant biological processes. To investigate the significance of our findings, we selected *PPP6R3*, the gene with the strongest support from our analysis which was not previously implicated in the regulation of BMD, for further investigation. We observed that *Ppp6r3* deletion in mice decreased BMD. In this work, we provide an updated resource of putatively causal BMD genes and demonstrate that *PPP6R3* is a putatively causal BMD GWAS gene. These data increase our understanding of the genetics of BMD and provide further evidence for the utility of combined TWAS/colocalization approaches in untangling the genetics of complex traits.

## Editor's evaluation

Many GWAS studies have been done to understand the genetic contributions to bone density, but very few have managed to pinpoint the gene affected by a polymorphism that caused an observed difference. In this paper, your team shows how scientists can identify causative variants from GWAS studies.

## Introduction

Osteoporosis, a disease characterized by low bone mineral density (BMD), decreased bone strength, and an increased risk of fracture, affects over 10 million individuals in the United States (*Black and Rosen, 2016*; *Burge et al., 2007*). BMD is the single strongest predictor of fracture and a highly heritable quantitative trait (*Miller et al., 1999*; *Ralston and Uitterlinden, 2010*; *Peacock et al., 2002*). Over the last decade, genome-wide association studies (GWASs) have identified over 1100 independent associations for BMD (*Morris et al., 2019*; *Estrada et al., 2012*; *Kemp et al., 2017*). However, despite the success of GWAS, few of the underlying causal genes have been identified (*Rocha-Braz and Ferraz-de-Souza, 2016*; *Sabik and Farber, 2017*).

One of the main difficulties in GWAS gene discovery is that the vast majority (>90%) of associations are driven by non-coding variation (*Giral et al., 2018*; *Edwards et al., 2013*). Over the last decade, approaches such as transcriptome-wide association studies (TWASs) and expression quantitative trait locus (eQTL) colocalization have been developed which leverage transcriptomic data in order to inform gene discovery by connecting non-coding disease-associated variants to changes in transcript levels (*Gusev et al., 2016*; *Barbeira et al., 2018*; *Pividori et al., 2020*; *Giambartolomei et al., 2014*; *Wen et al., 2017*). These approaches have proven successful for a wide array of diseases and disease-associated quantitative traits (*Pividori et al., 2020*; *Bhattacharya et al., 2020*; *Thom and Voight, 2020*). However, the osteoporosis field has lagged behind such efforts, due to the limited number of large-scale bone-related transcriptomic datasets.

In a TWAS, genetic predictors of gene expression (e.g., local eQTL – sets of genetic variants that influence the expression of a gene in close proximity, *Nica and Dermitzakis, 2013*) identified in a reference population (e.g., the Genotype-Tissue Expression [GTEx] project, *Consortium, 2013*) are used to impute gene expression in a GWAS cohort. Components of gene expression due to genetic variation are then associated with a disease or disease-associated quantitative trait. Genes identified by TWAS are often located in GWAS associations, suggesting that the genetic regulation of their expression is the mechanism underlying such associations. Several tools, for example, FUSION, PrediXcan, and MultiXcan (*Gusev et al., 2016*; *Gamazon et al., 2015*; *Barbeira et al., 2019*) have been developed to perform TWASs. Most of these tools use GWAS summary statistics, making TWAS widely applicable to large GWAS datasets. In contrast, eQTL colocalization is a statistical approach that determines if there is a shared genetic basis for two associations (e.g., a local eQTL and BMD GWAS locus). Recently, it has been demonstrated that prioritizing genes using both TWAS and eQTL colocalization provides a way to identify genes with the strongest support for causality (*Barbeira et al., 2018*; *Pividori et al., 2020*).

The GTEx project has generated RNA-seq data on over 50 tissues across hundreds of individuals (*Consortium, 2020*). Even though data on the tissues/cell types likely to be most relevant to BMD (bone or bone cells) were not included, the project demonstrated that many eQTL were shared across tissues (*Consortium, 2020*; *Battle et al., 2017*). Additionally, it is well known that effects in a wide-array of non-bone cell types and tissues can impact bone and BMD (*Fitzpatrick, 2002*; *Mirza and Canalis, 2015*). As a result, we sought to use the GTEx resource in conjunction with TWAS and eQTL colocalization to leverage non-bone gene expression data to identify putatively causal genes underlying BMD GWAS.

Here, we performed TWAS and eQTL colocalization using the GTEx resource and the largest BMD GWAS performed to date to identify potentially causal genes (*Morris et al., 2019*). Using this approach, we identified 512 genes significantly associated via TWAS with a significant colocalizing eQTL. To investigate the significance of our findings we selected Protein Phosphatase 6 Regulatory Subunit 3 (*PPP6R3*), the gene with the strongest support not previously implicated in the regulation of BMD, for further investigation. We demonstrate using mutant mice that *Ppp6r3* is a regulator of lumbar spine BMD. These results highlight the power of leveraging GTEx data, even in the absence

of data from the most relevant tissue/cell types, to increase our understanding of the genetic architecture of BMD.

## Results

### TWAS and eQTL colocalization identify potentially causal BMD GWAS genes

To identify potentially causal genes responsible for BMD GWAS associations, we combined TWAS and eQTL colocalization using GTEx data (*Figure 1A*). We began by performing a TWAS using reference gene expression predictions from GTEx (Version 8; 49 tissues) and the largest GWAS performed to date for heel estimated BMD (eBMD) (>1100 independent associations) (*Morris et al., 2019*; *Consortium, 2020*). The analysis was performed using S-MultiXcan, which allowed us to leverage information across all 49 GTEx tissues (*Barbeira et al., 2019*). Our analysis focused on protein-coding genes (excluded non-coding genes). A total of 2156 protein-coding genes were significantly (Bonferroni-adjusted p value ≤0.05) associated with eBMD (*Supplementary file 1a*).

Next, we identified colocalizing eQTL from each of the 49 tissues in GTEx using fastENLOC (*Pividori et al., 2020*; *Wen et al., 2017*). We identified 1182 colocalizing protein-coding genes with a regional colocalization probability (RCP) of 0.1 or greater (*Supplementary file 1b*). In total, 512 protein-coding genes were significant in both the TWAS and eQTL colocalization analyses (*Table 1* and *Supplementary file 1c*). Among the identified genes were many with well-known roles in the regulation of BMD, such as *RUNX2* (*Figure 1B*), *IGF1*, and *LRP6*, as well as novel genes such as *RERE* (*Figure 1C*). Overall, the identified genes had significantly colocalizing eQTL in all 49 GTEx tissues, with eQTL from cultured fibroblasts (132 genes), subcutaneous adipose tissue (117 genes), tibial artery (115 genes), and tibial nerve (114 genes) exhibiting the highest number of significant colocalizations (*Supplementary file 1d*). TWAS predictors were only generated for genes on autosomes and of the 1103 independent associations identified by *Morris et al., 2019*, 1097 were autosomal. For each of these, we defined a locus as the region consisting of ±1 Mbp around each association. Of the 1097 loci, almost half (542; 49%) of the loci contained at least one of the 512 prioritized genes. Most loci overlapped one gene (mean = 1.7, median = 1); however, 184 loci overlapped multiple genes, including a locus on Chromosome (Chr.) 20 (lead SNP rs6142137) which contained 9 prioritized genes (*Figure 1D* and *Figure 1—figure supplement 1*).

### Characterization of genes identified by TWAS/eQTL colocalization

To evaluate the ability of the combined TWAS/colocalization approach to identify genes previously implicated in the regulation of BMD, other bone traits, or the activity of bone cells, we queried the presence of 'known bone genes' within the list of the 512 prioritized protein-coding genes. To do so, we created a database-curated set of genes previously implicated in the regulation of bone processes (henceforth referred to as our 'known bone genes' list, N = 1399, *Supplementary file 1e*). Of the 512 genes identified above, 66 (12.9%) were known bone genes, representing a significant enrichment (odds ratio [OR] = 1.72; p = $1.0 \times 10^{-4}$) over what would be expected by chance (*Supplementary file 1f*).

We also performed a Gene Ontology (GO) enrichment analysis of the 512 prioritized genes. We observed enrichments in several bone-relevant ontologies, such as 'ossification' (p = $3.1 \times 10^{-6}$), 'skeletal system development' (p = $2 \times 10^{-5}$), and 'regulation of osteoblast differentiation' (p = $3.9 \times 10^{-5}$) (*Figure 2A* and *Supplementary file 1g*).

To compare our approach with the approach of prioritizing the closest genes to GWAS associations as potentially causal, we quantified the number of genes that were both the closest genes to eBMD GWAS associations and were members of the 'known bone gene' list. Of the 863 genes that were the closest genes to eBMD GWAS associations (*Supplementary file 1h*), 139 were members of the 'known bone gene list', representing a more significant enrichment of 'known bone genes' than our prioritization approach (OR = 2.56, p = $6.37 \times 10^{-19}$). Of our 512 prioritized genes, 206 (40%) were also the closest genes to eBMD GWAS associations, with 27 of the remaining 306 prioritized genes (8.8%) being members of the 'known bone gene' list.

The International Mouse Phenotype Consortium (IMPC) has recently measured whole body BMD in hundreds of mouse knockouts (*Dickinson et al., 2016*; *Swan et al., 2020*). We searched the IMPC

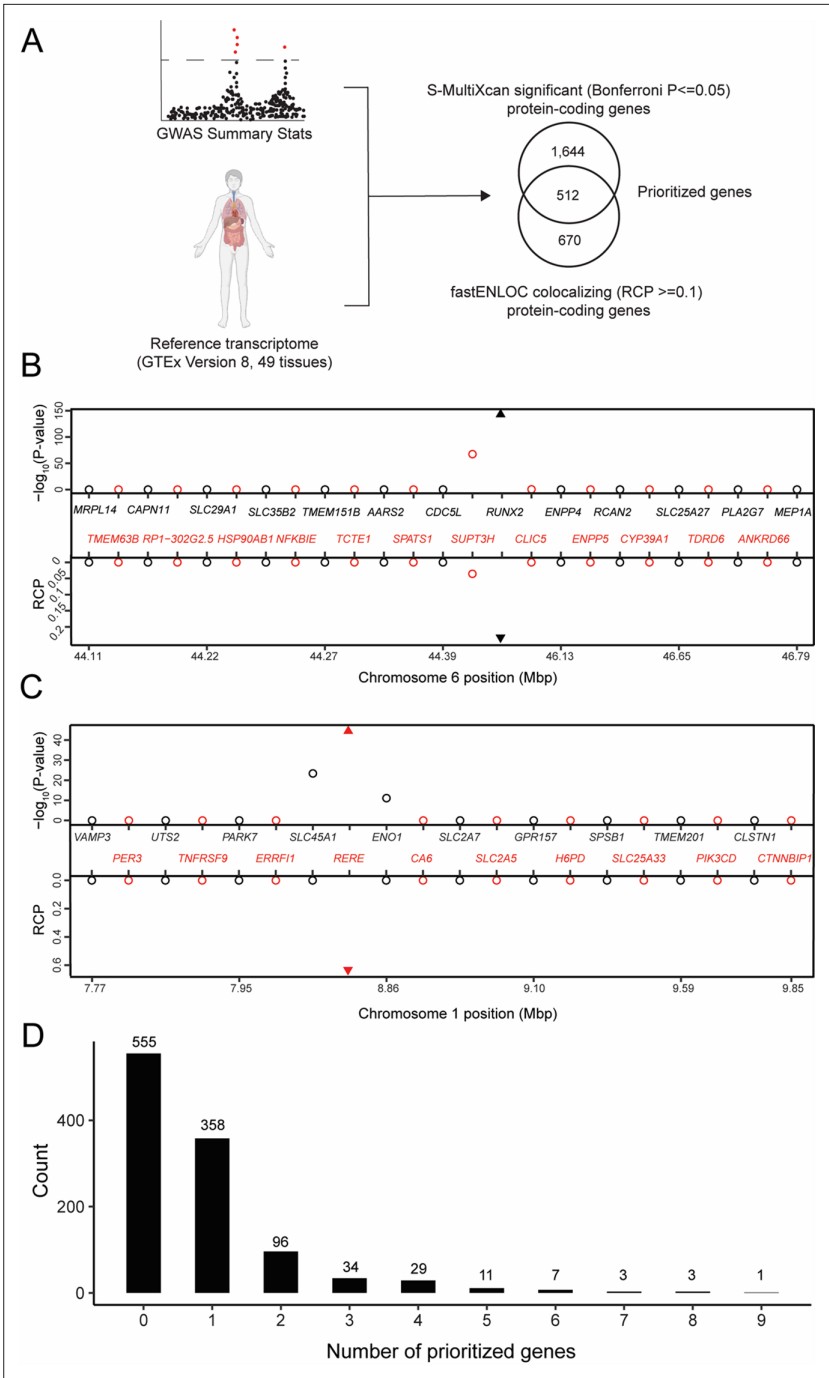

**Figure 1.** Transcriptome-wide association study (TWAS) and expression quantitative trait loci (eQTL) colocalization identify potentially causal bone mineral density (BMD) genome-wide association study (GWAS) genes. (**A**) Overview of the analysis. The human image was obtained from BioRender.com. TWAS/colocalization plot for genes in the locus around *RUNX2* (**B**) and *RERE* (**C**). The −log10 Bonferroni-adjusted p values from the TWAS analysis (top panel) and the maximum regional colocalization probabilities (RCPs) from the colocalization analyses (bottom panel). Genes alternate in color for visual clarity. Triangles represent *RUNX2* (**B**) and *RERE* (**C**). (**D**) Distribution of prioritized genes within estimated BMD (eBMD) GWAS loci.

The online version of this article includes the following figure supplement(s) for figure 1:

**Figure supplement 1.** Chromosome 20 contains a locus with nine prioritzed genes.

**Table 1.** Top 10 protein-coding genes significant by colocalization (RCP ≥0.1) and TWAS, sorted by TWAS p value.

| Gene | Tissue with greatest RCP | Max. RCP | TWAS p value (Bonferroni) |
|---|---|---|---|
| SPTBN1 | Cells_Cultured_Fibroblasts | 0.9469 | $<5 \times 10^{-324}$ |
| CCDC170 | Spleen | 0.6582 | $<5 \times 10^{-324}$ |
| FAM3C | Artery_Tibial | 0.4917 | $<5 \times 10^{-324}$ |
| SEPT5 | Skin_Sun_Exposed | 0.4868 | $2.26 \times 10^{-286}$ |
| FGFRL1 | Cells_Cultured_Fibroblasts | 0.1611 | $5.31 \times 10^{-272}$ |
| GREM2 | Cells_Cultured_Fibroblasts | 0.9998 | $4.31 \times 10^{-257}$ |
| GPATCH1 | Whole_Blood | 0.3564 | $3.44 \times 10^{-226}$ |
| RHPN2 | Pituitary | 0.2181 | $8.71 \times 10^{-221}$ |
| BMP4 | Brain_Cortex | 0.5468 | $5.49 \times 10^{-169}$ |
| RUNX2 | Esophagus_Gastroesophageal_Junction | 0.2372 | $1.99 \times 10^{-146}$ |

database for any of the 512 genes identified by TWAS and eQTL colocalization. Of the 512, 142 (27.7%) had been tested by the IMPC and 64 (12.5% of the 512 prioritized genes, 45% of the 142 IMPC-tested genes) had a nominally significant (p ≤ 0.05) alteration of whole-body BMD in knockout/ knockdown mice, compared to controls. Of the 64, 49 (76.5%) were not members of the 'known bone gene' list (*Figure 2—figure supplement 1A, B*).

An example of one of the 64 genes is *GPATCH1*, located within a GWAS association on human chromosome *19q13.11*. *GPATCH1* is part of the catalytic step 2 spliceosome, and may be involved in mRNA splicing, and is predicted to enable RNA-binding activity. It is also expressed in bone cells in mouse (*Alliance of genome resources, 2022*; *Lattin et al., 2008*). Of all the genes in the region, *GPATCH1* had the strongest TWAS association (p = $3.44 \times 10^{-226}$) (*Figure 2B*) and the strongest eQTL colocalization (whole blood, RCP = 0.36) (*Figure 2B–D*). The eQTL and BMD GWAS allele effects for the top SNPs were in the same direction, suggesting that decreasing the expression of *GPATCH1* would lead to decreased BMD. BMD data from the IMPC showed that female mice heterozygous for a *Gpatch1* null allele had decreased BMD (p = $2.17 \times 10^{-8}$) (*Figure 2E*). Together, these data suggest that many of the genes identified by the combined TWAS/colocalization approach are likely causal BMD GWAS genes.

## *PPP6R3* is a candidate causal gene for a GWAS association on Chr. 11

To identify novel candidate genes for functional validation, we focused on genes with the strongest evidence of being causal. To do so, we increased the colocalization RCP threshold to 0.5, and then sorted genes based on TWAS Bonferroni-adjusted p values. Furthermore, we constrained the list of candidates for functional validation to genes which were not members of the 'known bone gene' list or genes with a nominal (p ≤ 0.05) alteration in whole-body BMD as determined by the IMPC. This yielded 137 putatively causal BMD genes (*Table 2*, *Supplementary file 1i*, and *Figure 2—figure supplement 1C*).

Though it was not on the 'known bone gene' list, the first gene ranked by TWAS p value, *SPTBN1*, has been demonstrated to play a role in the regulation of BMD (*Calabrese et al., 2017*). The second, *PPP6R3*, has not been previously implicated in the regulation of BMD. *PPP6R3,* a regulatory subunit of protein phosphatase 6, which shows ubiquitous expression across tissues in humans (*Cristiano, 2020*), is located on human Chr. 11 within 1 Mbp of seven independent eBMD GWAS SNPs identified by *Morris et al., 2019* (subsequently referred to as 'eBMD lead SNPs') (*Figure 3A*). Of all the protein-coding genes (*N* = 29) in the ~1.8 Mbp region surrounding *PPP6R3*, its expression was the most significantly associated with eBMD by TWAS (Bonferroni = $5.7 \times 10^{-93}$) (*Figure 3B*). Furthermore, *PPP6R3* was the only gene in the region with eQTL (in four GTEx tissues, thyroid, ovary, brain_putamen_basal_ ganglia, and stomach with RCPs = 0.53, 0.50, 0.28, and 0.14, respectively) that colocalized with at least one of the eBMD associations (*Figure 3B*). Based on these data, we chose to further investigate *PPP6R3* as a potentially causal BMD gene.

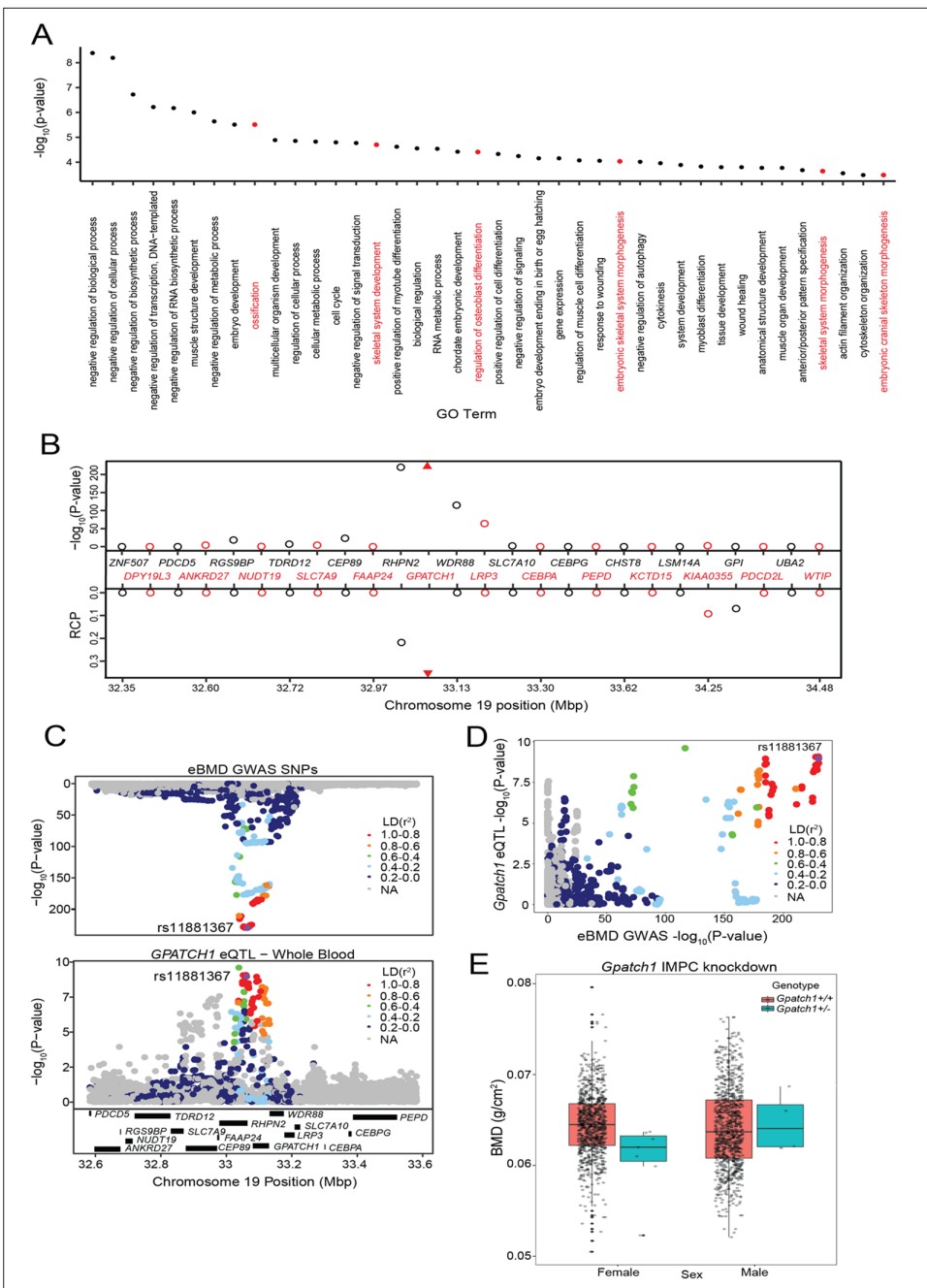

**Figure 2.** Transcriptome-wide association study (TWAS) and expression quantitative trait loci (eQTL) colocalization identify *Gpatch1* a potentially causal bone mineral density (BMD) genome-wide association study (GWAS) gene. (**A**) The top 40 terms from a Gene Ontology analysis of the 512 potentially causal BMD genes identified by our analysis. Terms with clear relevance to bone are highlighted in red. Only terms from the 'Biological Process' (BP) subontology are listed, and similar terms were removed for clarity. (**B**) TWAS/colocalization plot for genes in the locus around *GPATCH1* (±1.5 Mbp). The −log$_{10}$ Bonferroni-adjusted p values from the TWAS analysis (top panel) and the maximum regional colocalization probabilities (RCPs) from the colocalization analyses (bottom panel). Genes alternate in color for visual clarity. Triangles represent *GPATCH1*. (**C**) Mirrorplot showing estimated BMD (eBMD) GWAS locus (top panel) and colocalizing *GPATCH1* eQTL in whole blood (bottom panel). SNPs are colored by their linkage disequilibrium (LD) with rs11881367 (purple), the most significant GWAS SNP in the locus. (**D**) Scatterplot of −log$_{10}$ p values for *GPATCH1* eQTL versus eBMD GWAS SNPs. SNPs are colored by their LD with rs11881367 (purple). (**E**) BMD in *Gpatch1* knockdown mice. N = 7 females and N = 4 males for *Gpatch1*$^{+/−}$ mice, N

*Figure 2 continued on next page*

*Figure 2 continued*

= 880 females and *N* = 906 males for *Gpatch1*$^{+/+}$ mice. Boxplots indicate the median (middle line), the 25th and 75th percentiles (box) and the whiskers extend to 1.5 * IQR.

The online version of this article includes the following figure supplement(s) for figure 2:

**Figure supplement 1.** Schematic of prioritization pipeline.

We first determined which of the seven associations colocalized with the *PPP6R3* eQTL (*Figure 3C*). The most significant *PPP6R3* eQTL SNP in thyroid tissue (the tissue with the highest RCP) was rs10047483 (Chr. 11, 68.464237 Mbp) (*PPP6R3* eQTL p = 6.99 × 10$^{-8}$, eBMD GWAS p = 1.2 × 10$^{-100}$) located in intron 1 of *PPP6R3*. The most significant eBMD lead SNP in the locus was rs11228240 (Chr. 11, 68.450822 Mbp, eBMD GWAS p = 6.6 × 10$^{-101}$, *PPP6R3* eQTL p = 3.7 × 10$^{-6}$), located upstream of *PPP6R3*. Consistent with the colocalization analysis, these two variants are in high LD ($r^2$ = 0.941) and rs10047483 does not exhibit strong LD ($r^2$ < 0.104) with any of the other six eBMD lead SNPs in the locus. The eQTL and BMD GWAS allele effects for rs10047483 were opposing, suggesting that a decrease in the expression of *PPP6R3* would lead to an increase in BMD.

A recent fracture GWAS identified 14 significant associations, one of which was located in the *PPP6R3* region (rs35989399, Chr. 11, 68.622433 Mbp) (*Morris et al., 2019*). We analyzed the fracture GWAS in the same manner as we did above for eBMD. We found that *PPP6R3* expression when analyzed by TWAS was significant for fracture (TWAS Bonferroni-pval = 6.0 × 10$^{-3}$) and the same *PPP6R3* eQTL colocalized with the fracture association (RCP = 0.49 in ovary, RCP = 0.36 in thyroid) (*Figure 3D*). Together, these data highlight *PPP6R3* as a strong candidate for one of the seven eBMD/fracture associations in this region.

## *PPP6R3* is a regulator of femoral geometry, BMD, and vertebral microarchitecture

To assess the effects of *PPP6R3* expression on bone phenotypes, we characterized mice harboring a gene-trap allele (*Ppp6r3*$^{tm1a(KOMP)Wtsi}$) (*Figure 4A*). We intercrossed mice heterozygous for the mutant allele to generate mice of all three genotypes (wild-type (WT, *N* = 26 females, 21 males), heterozygous (HET, *N* = 33 females, 36 males), and mutant (MUT, *N* = 20 females, 21 males)). The absence of PPP6R3 protein in MUT mice was confirmed through western blotting (*Figure 4B*). Furthermore, we sequenced RNA extracted from the L5 vertebrae of wild-type and mutant mice, and found that *PPP6R3* was differentially expressed between the mutant and wild-type mice (p$_{adj}$ = 3.67 × 10$^{-104}$, log$_2$ fold change = −1.63). We did not observe changes in gene expression for any other gene in the locus, including *Lrp5* (p$_{adj}$ = 0.99, log$_2$ fold change = −0.068), a well-known regulator of bone mass just upstream of *Ppp6r3* (*Supplementary file 1j*).

**Table 2.** Top 10 novel protein-coding genes significant by colocalization (RCP ≥0.5) and TWAS, sorted by TWAS p value.

| Gene | Tissue with greatest RCP | Max. RCP | TWAS p value (Bonferroni) | # GTEx tissues with RCP ≥0.5 |
|---|---|---|---|---|
| *SPTBN1* | Cells_Cultured_fibroblasts | 0.9469 | <5 × 10$^{-324}$ | 2 |
| *PPP6R3* | Thyroid | 0.5291 | 5.7 × 10$^{-93}$ | 1 |
| *BARX1* | Colon_Transverse | 0.7764 | 6.36 × 10$^{-63}$ | 1 |
| *MEOX2* | Brain_Nucleus_accumbens_basal_ganglia | 0.6286 | 3.21 × 10$^{-53}$ | 3 |
| *RERE* | Adipose_Subcutaneous | 0.6431 | 6.95 × 10$^{-46}$ | 4 |
| *SIPA1* | Nerve_Tibial | 0.9981 | 4.26 × 10$^{-41}$ | 1 |
| *CAPZB* | Testis | 0.6716 | 3.64 × 10$^{-33}$ | 1 |
| *B4GALNT3* | Artery_Aorta | 0.9241 | 2.67 × 10$^{-33}$ | 4 |
| *TRPC4AP* | Breast_Mammary_Tissue | 0.5577 | 8.62 × 10$^{-31}$ | 3 |
| *AXL* | Minor_Salivary_Gland | 0.6205 | 9.74 × 10$^{-31}$ | 3 |

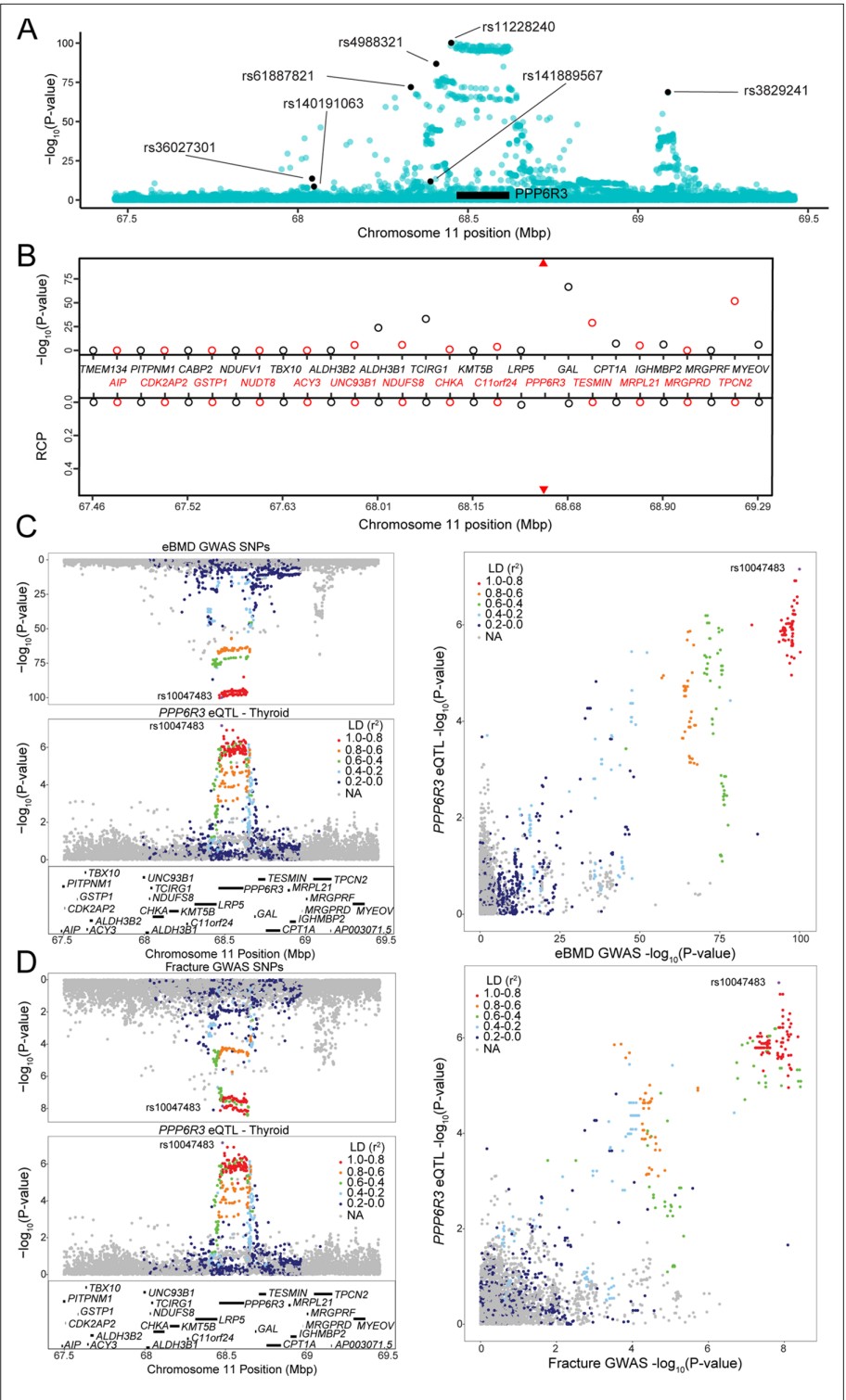

**Figure 3.** PPP6R3 is a top 10 novel estimated bone mineral density (eBMD) gene. (**A**) eBMD genome-wide association study (GWAS) SNPs around the *PPP6R3* locus (±1 Mbp). The *y*-axis represents −log₁₀ eBMD GWAS p values. Highlighted SNPs (black) are the seven lead eBMD GWAS SNPs in the locus. (**B**) Transcriptome-wide association study (TWAS)/colocalization plot for genes in the locus around *PPP6R3* (±1 Mbp). The −log₁₀ Bonferroni-adjusted p values from the TWAS analysis (top panel) and the maximum regional colocalization probabilities (RCPs) from the colocalization analyses (bottom panel). Genes alternate in color for visual clarity. Triangles represent *PPP6R3*. Mirrorplot of the eBMD locus (**C**) and *PPP6R3* expression quantitative trait loci (eQTL)

*Figure 3 continued on next page*

*Figure 3 continued*

in thyroid, and fracture locus and *PPP6R3* eQTL in thyroid (**D**). The panels on the right are scatterplots of $-\log_{10}$ p values for *PPP6R3* eQTL and eBMD GWAS SNPs (**C**) and the *PPP6R3* eQTL and fracture GWAS SNPs (**D**). SNPs are colored by their linkage disequilibrium (LD) with rs10047483 (purple), the most significant *PPP6R3* eQTL in the locus. Not all genes are shown.

The BMD analyses presented above used heel eBMD GWAS data. We used these data because they represent the largest, most well-powered BMD GWAS to date (*Estrada et al., 2012*). However, to determine whether perturbation of *Ppp6r3* would be expected to impact femoral or lumbar spine BMD in a similar manner, we turned to a smaller GWAS to look at both of these traits. In a GWAS by *Estrada et al., 2012*, a total of 56 loci were identified for femoral neck (FNBMD) and lumbar spine (LSBMD) BMD. One of the 56 loci corresponded to the same SNPs associated with the *PPP6R3* eQTL. The locus was significant for LSBMD; however, it did not reach genome-wide significance for FNBMD (*Figure 4—figure supplement 1*).

We evaluated BMD at both the femur and the lumbar spine in *Ppp6r3*^tm1a(KOMP)Wtsi^ mice, with the expectation, based on the above data, that perturbation of *Ppp6r3* would have a stronger impact on BMD at the lumbar spine. At approximately 9 weeks of age, we measured areal BMD (aBMD) at the femur and lumbar spine using dual X-ray absorptiometry (DXA). First, we observed no change in body weight at 9 weeks that might impact bone phenotypes (*Figure 4—figure supplement 2A*). As the above analysis predicted, we observed a significant effect of *Ppp6r3* genotype on aBMD at the lumbar spine (WT vs. MUT p = 0.01, *Figure 4—figure supplement 2C*), but not the femur (WT vs. MUT p = 0.26, *Figure 4—figure supplement 2D*). It should also be noted, however, that we observed significantly decreased femoral width, but not length, in *Ppp6r3* mutant mice (anterior–posterior [AP] femoral width, WT vs. MUT p = 0.02; medial–lateral [ML] femoral width, WT vs. MUT p = $2.2 \times 10^{-6}$, *Figure 4—figure supplement 2B–D*).

We further characterized the effects of *Ppp6r3* genotype on microarchitectural phenotypes, in spine and femur, via micro-computed tomography (μCT). We observed significant (p ≤ 0.05) decreases in trabecular bone volume fraction (BV/TV, WT vs. MUT p = 0.015, *Figure 4—figure supplement 2E, F*) and volumetric BMD (vBMD, WT vs. MUT p = 0.015, *Figure 4—figure supplement 2G*) of the lumbar spine as a function of *Ppp6r3* genotype, but found no significant changes in tissue mineral density (TMD, *Figure 4—figure supplement 2E*), trabecular separation (TbSp), trabecular thickness (TbTh), or trabecular number (TbN) (*Figure 4—figure supplement 2F–H*). In the femoral midshaft, we observed significant decreases in total area (Tot.Ar, WT vs. MUT p = 0.00724, *Figure 4—figure supplement 2H*) and medullary area (Ma.Ar, WT vs. MUT p = 0.00903, *Figure 4—figure supplement 2I*), which are concordant with the aforementioned observed decreases in femoral width in *Ppp6r3* mutant mice. We did not observe any changes in trabecular bone parameters in the distal femur.

We also measured Procollagen 1 Intact N-Terminal Propeptide P1NP levels in plasma, and observed a significant increase in P1NP levels in mutant mice (p = 0.00422), suggesting that the observed decrease in bone mass is due to increased bone turnover (*Figure 4—figure supplement 2I–K*). Finally, to assess the effects of *Ppp6r3* genotype on bone matrix composition, we performed periosteal Raman spectroscopy on both the lumbar spines and femurs. We did not observe any significant (p ≤ 0.05) effects of *Ppp6r3* genotype on bone matrix composition (*Figure 4—figure supplements 3–6*).

## Discussion

BMD GWASs have identified over 1100 associations to date. However, identifying causal genes remains a challenge. To aid researchers in further dissecting the genetics of complex traits, reference transcriptomic datasets and computational methods have been developed for the prioritization and identification of causal genes underlying GWAS associations. In this work, our goal was to utilize these data and tools to prioritize putatively causal genes underlying BMD GWAS associations. Specifically, we used the GTEx eQTL reference dataset in 49 tissues to perform TWAS and eQTL colocalization on the largest BMD GWAS. Using this approach, we identified 512 putatively causal protein-coding genes that were significant in both the TWAS and colocalization approaches.

Our approach was inspired by a recent study that used the GTEx resource and a TWAS/eQTL colocalization approach similar to the one we employed. *Pividori et al., 2020* recently combined

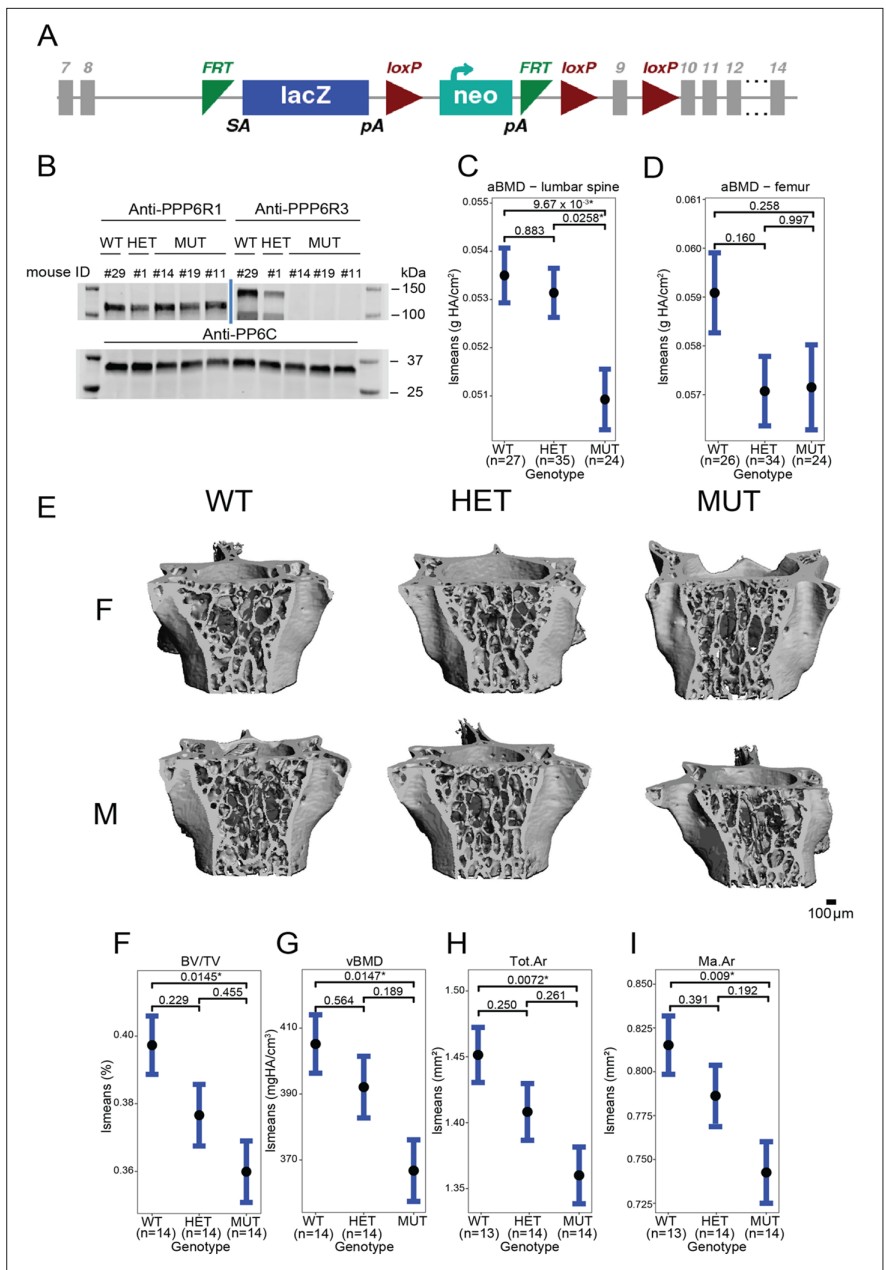

**Figure 4.** Ppp6r3 functional validation shows an effect of genotype on bone mass. (**A**) Schematic of the *Ppp6r3* gene-trap allele (Ppp6r3tm1a(KOMP)Wtsi). Image obtained from the International Mouse Phenotype Consortium (IMPC). (**B**) Western blot of *Ppp6r3* in experimental mouse spleens. Top left panel shows that PPP6R1 protein (control) levels are not affected by the *Ppp6r3* gene-trap allele. Top right panel shows the effect of the gene-trap allele on PPP6R3 protein levels. The two bands are ostensibly due to different PPP6R3 isoforms. Bottom panel shows that PP6C protein (control) levels are not affected by the *Ppp6r3* gene-trap allele. For raw blots, refer to *Figure 4—source data 1*. Least-squares means for spinal (**C**) and femoral (**D**) areal BMD (aBMD) dual X-ray absorptiometry (DXA) in *Ppp6r3* wild-type (WT), heterozygous (HET), and mutant (MUT) mice. Contrast p values, adjusted for multiple comparisons are presented. *p ≤ 0.05. (**E**) Representative images of vertebrae for the *Ppp6r3* experimental mice. Scale is shown on the bottom right. (**F, G**) Least-squares means for micro-computed tomography (µCT) measurements in the lumbar spines of *Ppp6r3* WT, HET, and MUT mice. Contrast p values, adjusted for multiple comparisons are presented. *p ≤ 0.05. (**H, I**) Least-squares means for µCT measurements in the femoral midshaft of *Ppp6r3* WT, HET, and MUT mice. Contrast p values, adjusted for multiple comparisons are presented. *p ≤ 0.05. Abbreviations: BV/TV – bone volume fraction, vBMD – volumetric bone mineral density, Tot.Ar – total area, Ma.Ar – medullary area. In panels (**C, D and F–I**), the center points represent the least-square

*Figure 4 continued on next page*

*Figure 4 continued*

means, and the error bars represent confidence intervals at a confidence level of 0.9. Data presented here are from both male and female mice.

The online version of this article includes the following source data and figure supplement(s) for figure 4:

**Source data 1.** Raw, unedited, PPP6R3 western blots.

**Figure supplement 1.** LSBMD and FNBMD genome-wide association study (GWAS) SNPs in the PPP6R3 locus.

**Figure supplement 2.** *Ppp6r3* functional validation.

**Figure supplement 3.** Raman spectroscopy in femur, means.

**Figure supplement 4.** Raman spectroscopy in spines, means.

**Figure supplement 5.** Raman spectroscopy in femurs, standard deviation.

**Figure supplement 6.** Raman spectroscopy in spines, standard deviation.

TWAS and eQTL colocalization to GTEx and GWAS data on 4091 traits, including BMD, from the UK Biobank data. A total of 76 protein-coding genes were identified and of the 76, we identified 55 (72.4%) of the same genes in our implementation. There are several reasons for this discrepancy in the number of prioritized genes. First, both studies used a GWAS based on the UK Biobank (**Bycroft et al., 2018**); however, sample sizes were different. The PhenomeXcan project utilized GWAS data based on the analysis of ~207,000 individuals, whereas we used GWAS data based on the analysis of ~426,000 individuals (**Morris et al., 2019**; **Pividori et al., 2020**). Second, the two GWAS studies utilized different association models. Finally, due to the breadth of the PhenomeXcan project, they had a higher multiple-testing burden than we did, which led to different Bonferroni-adjusted p value thresholds (p < $5.49 \times 10^{-10}$ vs. p $\leq 2.38 \times 10^{-6}$).

One of many novel genes identified in our study was *PPP6R3*, which was also identified in the PhenomeXcan project (**Pividori et al., 2020**). *PPP6R3* is a regulatory subunit of protein phosphatase 6 and has been implicated in several cancers (**Cristiano, 2020**; **Stefansson and Brautigan, 2006**). In humans, the PPP6R3 protein shows ubiquitous expression across tissues, and may have an important role in maintaining immune self-tolerance (**Cristiano, 2020**). It is unclear how *PPP6R3* may be influencing BMD. However, protein phosphatase 6 has been shown to oppose activation of the nuclear factor kappa-light-chain enhancer of activated B cells (NF-κB) pathway in lymphocytes (**Ziembik et al., 2017**). Since the NF-κB signaling pathway is highly involved in osteoclastogenesis and bone resorption, it is possible that *PPP6R3* may be involved in the regulation of this pathway in osteoclasts (**Abu-Amer, 2013**). Further studies that characterize the role of *PPP6R3*, and the effects of its deletion, in bone cells are required to further elucidate its effect on BMD.

The *PPP6R3* locus demonstrated a high level of complexity, containing seven independent GWAS associations, at least one of which was also associated with fracture. Interestingly, just upstream of *PPP6R3* is *LRP5*, a WNT signaling co-receptor (**Mao et al., 2001**). *LRP5* is a well-known regulator of BMD and gain and loss of function mutations lead to high bone mass syndrome and osteoporosis pseudoglioma, respectively (**Mizuguchi et al., 2004**; **Boyden et al., 2002**; **Marques-Pinheiro et al., 2010**; **Gong et al., 2001**). *LRP5* expression was not significantly associated with eBMD by TWAS (Bonferroni p = 1), nor did it have a colocalizing eQTL in GTEx tissues (most significant RCP = $1.6 \times 10^{-2}$ in pancreas). However, another eBMD lead SNP in the region, rs4988321, is a missense mutation in *LRP5* (Val667Met) that has been associated with BMD in multiple studies (**van Meurs et al., 2008**; **Brixen et al., 2007**; **Giroux et al., 2007**). While this variant represents an association that is independent of the rs10047483 association ($r^2$ = 0.104), it further highlights the complexity of this locus both in terms of the number of associations and target genes. We believe these data support a model of allelic heterogeneity involving multiple genes (at least *LRP5* and *PPP6R3*) at this locus, similar to other BMD loci identified by GWAS such as the 'Wnt16' locus (**Chesi et al., 2019**).

To determine the effect of *Ppp6r3* expression on bone, we characterized bone phenotypes in mice harboring a gene-trap allele (*Ppp6r3*^tm1a(KOMP)Wtsi^). Consistent with the observation that the *PPP6R3* eQTL SNPs were significantly associated with lumbar spine, but not femoral neck BMD, we observed that *Ppp6r3* deletion had a significant effect on lumbar spine BMD, but not femoral BMD. Using μCT, we further characterized the effect of *Ppp6r3* deletion on lumbar spine and femoral microarchitecture. We observed significant decreases in trabecular bone volume fraction (BV/TV) and vBMD of the lumbar spine as a function of *PPP6R3* genotype. While we did not observe significant effects of *Ppp6r3*

deletion on trabecular thickness or number, the direction of effects for those phenotypes suggests that the observed decrease in bone volume fraction and BMD may be explained by the cumulative but more subtle effects of *Ppp6r3* deletion on trabecular thickness and number. Consistent with our caliper-based observations that femoral width was decreased at the midshaft in *Ppp6r3* mutants, we observed a decrease in medullary and total area in the femoral midshaft. We also hypothesized that *Ppp6r3* deficiency might lead to defects in matrix composition; however, we did not observe any significant differences. The lack of differences might be due to the fact that we used relatively young mice for these assays.

Our hypothesis regarding the directions of effect of *Ppp6r3* expression on BMD based on the eQTL and eBMD/lumbar spine BMD GWAS were opposite to what we observed. There are several reasons that may explain this. First, our hypothesis was based on expression data in non-bone tissues and cell types. Recent studies have shown that the direction of eQTL effects can differ between different cells and tissues within humans (*Peters et al., 2016*; *Mizuno and Okada, 2019*). Second, our hypothesis was based on human data, while our functional experiments were performed in mice. Third, we globally deleted *Ppp6r3* in mice, as opposed to ablating it in a cell-type-specific manner. Future studies investigating which tissue/cell-type *Ppp6r3* is operative in and the generation of conditional *Ppp6r3* knockouts will allow us to unravel the precise role of this association and *Ppp6r3* in the regulation of bone mass.

As we and others have shown, the use of both TWAS and eQTL colocalization can prioritize putatively causal genes underlying GWAS associations. Here, we have shown the utility of this approach even in the absence of eQTL data from the most phenotype-relevant tissue. However, it is important to highlight the limitations of our analysis. While studies have shown that many eQTL are shared among tissues, the lack of eQTL data in bone and bone cells means that bone-specific eQTL were missed. For example, a study conducted by Mullin et al. performed eQTL colocalization and summary-based Mendelian randomization (SMR) by utilizing GWAS and expression data from osteoclast-like cells, and prioritized several eBMD genes (*Mullin et al., 2020*). Thirty-eight percent of the colocalizing eQTL and 19% of the SMR genes that they identified overlapped with our 512 prioritized genes, suggesting that we have missed many potential effector genes with eQTL specific to bone cells. In addition, the use of multiple non-bone tissues may have inflated the number of false positives based on coincidence of strong TWAS and eQTL colocalization signals that have no biological impact on bone. Furthermore, the lack of bone transcriptomic data may also explain the observed disparity between our hypothesized and observed direction-of-effect for *PPP6R3*. It is also important to note that due to the reliance of this approach on eQTL data, genes that affect BMD via non-expression-related mechanisms were not captured. Another limitation of our approach arises from the definition of loci based on linkage disequilibrium (LD). We used a set of previously defined approximately independent LD blocks, derived from a cohort of European individuals, in our fastENLOC analysis (*Berisa and Pickrell, 2016*). The inexact nature of these data may lead to spurious colocalizations due to mismatches in LD structure between the reference LD blocks and the GWAS/eQTL populations. Additionally, because the GWAS and eQTL data have mismatching LD structures, due to their being derived from cohorts with different ancestries, our analyses, particularly the colocalization analyses, may suffer from reduced power (*Hukku et al., 2021*). This also raises the related issue of the reduced generalizability of our results in non-European individuals, which brings further attention to the necessity of performing GWASs and providing reference data in diverse and underrepresented populations. Additionally, another issue arises when considering correlations in expression, and predicted expression, between genes in a locus, which may lead to spurious associations in TWAS analyses (*Wainberg et al., 2019*). Finally, as we show above, our method does not perform as well as prioritizing genes based on their proximity to GWAS associations. However, because our method utilizes systems genetics techniques and data, such as eQTL, we believe that our method prioritizes genes in a more biologically relevant manner. In fact, utilizing the closest gene method alone, *PPP6R3* would not have been prioritized as a bone-relevant gene. We suggest that future studies utilize both prioritization techniques, such as taking the closest genes to GWAS associations and cross referencing them with colocalizing and TWAS-associated genes, in order to provide further evidence for functional validation.

In summary, we applied a combined TWAS/colocalization approach using GTEx and identified 512 putatively causal BMD genes. We further investigated *PPP6R3* and demonstrated that it is a regulator of lumbar spine BMD. We believe this work provides a valuable resource for the bone genetics

community and may serve as a framework for prioritizing genes underlying GWAS associations using publicly available tools and data for a wide range of diseases.

## Methods

### fastENLOC colocalization

For each of the eBMD and fracture GWASs, we performed colocalization using fastENLOC, by following the tutorial and guidelines available at https://github.com/xqwen/fastenloc (*Wen, 2022*).

Briefly, for each GWAS, we converted variant coordinates to the hg38 human genome assembly, using the UCSC liftOver tool (minimum ratio of bases that must remap = 1; https://genome.ucsc. edu/cgi-bin/hgLiftOver). We calculated z-scores by dividing GWAS betas by standard errors. We then defined loci based on European LD blocks, as defined based on the results of *Berisa and Pickrell, 2016*.

Z-scores were then converted to posterior inclusion probabilities using torus (*Wen, 2015*). Finally, these data were colocalized with fastENLOC for all 49 GTEx V8 tissues, with the '-total_variants' flag set to 14,000,000. Colocalization was performed using pre-computed GTEx multi-tissue annotations, obtained from https://github.com/xqwen/fastenloc; *Wen, 2022*. Finally, to identify protein-coding genes in the results, we utilized Ensembl's 'hsapiens_gene_ensembl' dataset using biomaRt (version 2.45.8).

### S-MultiXcan

We conducted a TWAS by integrating genome-wide SNP-level association summary statistics from an eBMD GWAS (*Morris et al., 2019*) with GTEx version 8 gene expression QTL data from 49 tissue types. We used the S-MultiXcan approach for this analysis, to correlate gene expression across tissues to increase power and identify candidate susceptibility genes (*Barbeira et al., 2019*). Default parameters were used, with the exception of the '--cutoff_condition_number' parameter, which was set to 30. Bonferroni-correction of p values was performed on the resultant gene set (22,337 genes), using R's 'p.adjust' function. This was followed by the removal of non-protein-coding genes. The analysis was also performed in the same manner using summary statistics from a fracture GWAS (*Morris et al., 2019*). Finally, to identify protein-coding genes in the results, we utilized Ensembl's 'hsapiens_gene_ensembl' dataset using biomaRt (*Durinck et al., 2009*; *Durinck et al., 2005*).

### Creation of the 'known bone gene' list

We generated a 'known bone gene' set as follows: First, we downloaded Gene Ontology IDs for the following terms: 'osteo*', 'bone', and 'ossif*' from AmiGO2 (version 2.5.13) (*Carbon et al., 2009*). After removal of non-bone-related terms, we extracted all mouse and human genes related to the GO terms, using biomaRt. From this list, we retained protein-coding genes.

We also used the 'Human-Mouse: Disease Connection' database available at the Mouse Genome Informatics website, to download human and mouse genes annotated with the terms 'osteoporosis', 'bone mineral density', 'osteoblast', 'osteoclast', and 'osteocyte'. We used biomaRt to identify the gene biotypes, and retained protein-coding genes. We then used the MGI human-mouse homology table (http://www.informatics.jax.org/downloads/reports/HOM_MouseHumanSequence.rpt) to convert all mouse genes to their human homologs. Finally, we removed genes that were not interrogated in both the colocalization and the TWAS analyses.

### GO enrichment analyses

GO analysis was performed for the set of protein-coding genes passing the colocalization threshold RCP ≥0.1 and S-MultiXcan Bonferroni p value ≤0.05, using the 'topGO' package (version 2.46.0) in R (*Alexa and Rahnenfuhrer, 2021*). Enrichment tests were performed for the 'Molecular Function', 'Biological Process', and 'Cellular Component' ontologies, using all protein-coding genes that were subjected to colocalization and MultiXcan analysis as background. Enrichment was performed using the 'classic' algorithm with Fisher's exact test. p values were not adjusted for multiple testing.

## LD calculations

LD between variants was calculated using the LDlinkR (version 1.0.2) R package, using the 'EUR' population (*Myers et al., 2020*).

## *Ppp6r3* knockout mouse generation

The study was carried out in strict accordance with NIH's Guide for the Care and Use of Laboratory Animals. Additionally, the University of Virginia Institutional Animal Care and Use Committee approved all animal procedures. *Ppp6r3* gene-trap mice were generated using targeted embryonic stem cell clones heterozygous for the *Ppp6r3*[tm1a(KOMP)Wtsi] gene-trap allele obtained from the International Knockout Mouse Project (KOMP; https://www.komp.org). KOMP ES clones were karyotyped and injected using a XYClone Laser (Hamilton Thorne, Beverly, MA) into B6N-Tyr[c-Brd]/BrdCrCrl (Charles River, Wilmington, MA) eight-cell stage embryos to create chimeric mice. Resultant chimeras were bred to B6N-Tyr[c-Brd]/BrdCrCrl mice to obtain germline transmission of the *Ppp6r3* gene-trap allele. From a breeding pair of two heterozygous mice, we generated our experimental population through HET × HET matings. Breeder mice were fed a breeder chow diet (Envigo Teklad S-2335 mouse breeder sterilizable diet, irradiated. Product # 7904), and experimental mice were fed a standard chow diet (Envigo Teklad LM-485 irradiated mouse/rat sterilizable diet. Product #7912).

## Genotyping of *Ppp6r3* mice

DNA for genotyping was extracted from tail clips as follows: tail clips were incubated overnight at 55°C in a solution of 200 µl digestion/lysis buffer (Viagen Direct PCR [tail], Los Angeles, CA) and 1 mg/ml proteinase K (Viagen, Los Angeles, CA). After overnight incubation, tails were heated at 85°C for 45 min, and solutions were subsequently stored at 4°C.

For genotyping, PCR reactions were set up as follows. For each reaction, 1 µl of DNA was mixed with 24 µl of a master mix consisting of 19.5 µl nuclease-free $H_2O$, 2.5 µl 10× PCR reaction buffer (Invitrogen, Waltham, MA), 0.75 µl of $MgCl_2$ (Invitrogen, Waltham, MA), 0.5 µl of 10 mmol Quad dNTPs (Roche Diagnostics GmbH, Mannheim, Germany) 0.25 µl of Platinum Taq DNA polymerase (Invitrogen, Waltham, MA), and 0.25 µl of each primer, diluted to 20 µmol.

Primers: PCR primers were obtained from Integrated DNA Technologies, Coralville, IA.

> Forward primer: 5'-CAC CTG GGT TGG TTA CAT CC-3'
> Reverse primer: 5'-GAC CCT GCC TTA AAA CCA AA-3'

The following PCR settings were used:

- Initialization: 94°C, 120 s
- Denaturation: 94°C, 30 s (37 cycles)
- Annealing: 54°C, 30 s (37 cycles)
- Elongation: 72°C, 35 s (37 cycles)
- Final elongation: 72°C, 300 s

PCR products were run on a 2% agarose gel for 150 min at 60 V, to distinguish between wild-type, heterozygous and mutant *Ppp6r3* mice.

## PPP6R3 western blotting

Mouse spleens 20–40 mg in weight were suspended in 1% NP40 buffer (50 mM Tris [pH 8] 100 mM NaCl, 1% NP40, 1 mM EGTA(egtazic acid), 1 mM EDTA(ethylenediaminetetraacetic acid), Protease inhibitor cocktail (04-693-116-001, Roche), 1 mM PMSF(phenylmethylsulfonyl fluoride), 50 mM NaF, 0.2 mM sodium vanadate). The tissue was homogenized by RNase-free disposable pestles (Thermo Fisher #12-141-364) and incubated for 10 min on ice. After brief sonication, the sample was centrifuged for 10 min at 13,000 × rpm at 4°C. The protein concentration in the extract was measured by Bradford assay. 100 µg of sample protein was boiled 5 min in sodium dodecyl sulfate (SDS) sample buffer, loaded in each lane, resolved by gradient SDS–polyacrylamide gel electrophoresis (Bio-Rad #456-1085) and immunoblotted as described in *Guergnon et al., 2009*. Primary antibodies were diluted 1:1000 (SAPS1 Ab: Thermo Fisher #PA5-44275, SAPS3 Ab: Thermo Fisher #PA5-58405, PP6C Ab: Sigma #HPA050940).

### *PPP6R3* functional validation

Experimental mice of both sexes were sacrificed at approximately 9 weeks of age (mean age = 61 days). At sacrifice, the right femurs were isolated, and femoral morphology (length and widths in anterior–posterior and medial–lateral orientations) was measured with digital calipers (Mitoyuto American, Aurora, IL).

Femurs were then wrapped in phosphate-buffered saline (PBS)-soaked gauze and stored at −20°C, until analysis. Lumbar vertebrae L3–L5 were also dissected at sacrifice and were wrapped in PBS-soaked gauze and frozen at −20°C. Given our prior experience in measuring bone geometry and microstructure, we used a minimum $N$ = 14 for our analyses, ensuring 80% power to detect a statistically significant effect at an alpha ≤0.05.

### Bulk RNA isolation, sequencing, and quantification

We isolated RNA from a randomly chosen subset ($n$ = 16, 4/sex each of WT and MUT) of the available mice.Total RNA was isolated from L5 vertebrae, using the mirVana miRNA Isolation Kit (Life Technologies, Carlsbad, CA). Total RNA-seq libraries were constructed using Illumina TruSeq Stranded mRNA LT sample prep kits. Samples were sequenced to an average of 39 million 2 × 151 bp paired-end reads (total RNA-seq) on an Illumina NextSeq500 sequencer by Psomagen, Inc A custom bioinformatics pipeline was used to quantify RNA-seq data. Briefly, RNA-seq FASTQ files were quality controlled using FASTQC (version 0.11.5) and MultiQC (version 1.11). Reads were trimmed using bbduk (bbmap package version 38.57). Trimmed reads were then aligned to the mm10 genome assembly with HISAT2 (version 2.1), and quantified with Stringtie (version 1.3.3). Read count information was then extracted with a Python script provided by the Stringtie website (prepDE.py).

### Bulk RNA differential expression analyses

Using gene count matrices, differential expression was performed using DESeq2 (*Love et al., 2014*; Version 1.34.0) between wild-type and mutant samples. We used the 'DESeq' function from DESeq2, with a design formula of ~sex + genotype. p values were adjusted using the 'p.value' function, using the 'BH' method.

### Dual X-ray absorptiometry

Individual right femurs and the lumbar spine (L5 vertebrae) were isolated from surrounding soft tissues and frozen at −20°C in PBS. DXA was performed on the femurs and lumbar vertebrae using the Lunar Piximus II (GE Healthcare) as described previously by *Beamer et al., 2011*. In short, 10 isolated bones were placed in the detector field at a time and the samples were analyzed one by one, such that the region of interest (ROI) was set for one specimen at a time for data collection. The ROI for the femurs was on the entire isolated femur. For the spine, was on the entire isolated L5. Care was taken to ensure that the sample orientation was identical for all samples.

### μCT and image analysis

All μCT analyses were carried out at the μCT Imaging Core Facility at Boston University using a Scanco Medical μCT 40 instrument (Brütisellen, Switzerland). The power, current, and integration time used for all scans were 70 kVp, 113 μA, and 200 ms, respectively. The L5 vertebrae and right femora were scanned at a resolution of 12 μm/voxel. Two volumes of interest (VOIs) in the L5 were selected for analysis: (1) the entire portion of the L5 vertebra extending from 60 μm caudal to the cranial growth plate in the vertebral body to 60 μm cranial to the caudal growth plate; and (2) only the trabecular centrum contained in the first VOI. Semi-automated-edge detection (Scanco Medical) was used to define the boundary between the trabecular centrum and cortical shell to produce the second VOI. Two VOIs were also analyzed for each femur: (1) a 0.3-mm-long segment of the diaphysis, centered at the mid-point of the bone; and (2) a 1.2-mm-long segment of the distal metaphyseal trabecular compartment. To define the location of the latter, the location of the distal femoral growth plate was determined, and the distal end of the VOI was set at 60 μm proximal to that growth plate. Gaussian filtering (sigma = 0.8, support = 1) was used for partial background noise suppression. A scan of a potassium hydroxyapatite phantom allowed conversion of gray values to mineral density. For segmentation of bone tissue, the threshold was set at a 16-bit gray value of 7143 (521 mgHA/ccm), and this global threshold was applied to all of the samples. For each VOI, the following were calculated: total

volume (TV), bone volume (BV), bone volume fraction (BV/TV), BMD, and TMD. BMD was defined as the average density of all voxels in the VOI, whereas TMD was defined as the average density of all voxels in the VOI above the threshold (*Bouxsein et al., 2010*). For the second VOI, the following additional parameters were calculated: trabecular thickness (Tb.Th), trabecular separation (Tb.Sp), trabecular number (Tb.N), connectivity density (Conn.D), and structure model index (SMI) (*Bouxsein et al., 2010*).

For the femoral VOI, the following additional parameters were calculated instead: cortical thickness (Ct.Th), total area (Tot.Ar), marrow area (Ma.Ar), maximum and minimum moments of inertia, and polar moment of inertia (*Bouxsein et al., 2010*).

## P1NP collection and quantification

Plasma was collected via submandibular bleeding from isoflurane anesthetized wild-type and mutant *PPP6R3* mice (*N* = 10/sex/genotype). Plasma P1NP levels were measured using commercially available kits from IDS (Gaithersburg, MD), according to the manufacturer's instructions. The assay sensitivity was 0.7 ng/ml. The intra-assay variation was 6.3%, and the inter-assay variation was 8.5%. All measurements were performed in duplicate.

## Raman spectroscopy

Raman spectroscopy was performed using a Renishaw inVia Raman Microscope (Gloucestershire, UK) on each bone sample using a 785-nm-edge red incident laser. A rectangular filled map was created with 3 points in the *x*-axis and 20 points in the *y*-axis, for a total of 60 collected points. Each point was exposed 10 times for 6 s per exposure. A custom MATLAB script was used to evaluate the peak position, maximum intensity, peak width, full width at half maximum (FWHM), and the area under each peak. Peak area ratios were calculated for mineral:matrix, carbonate:phosphate, and crystallinity. Furthermore, the standard deviations of peak area ratios were calculated for each mouse, and were further used to evaluate the material heterogeneity in groups.

## Statistical analyses

To calculate the enrichment of bone genes in prioritized genes, we performed Fisher's exact test, using R's 'fisher.test' function, with the alternative hypothesis set as 'greater'.

For the statistical analysis of the phenotyping results, we calculated least-squares means (lsmeans) using the 'emmeans' R package (version 1.5.2.1) (*Lenth, 2020*). Input for the lsmeans function was a linear model including terms for genotype, weight, and age in days. For sex-combined data, we also added a term for sex. For DXA phenotypes, we included a term for 'CenterRectX' and 'CenterRectY'. For the Raman spectroscopy data, weight and age were not included as terms in the linear model.

We used Tukey's HSD(honest significance test) test to test for significant differences in lsmeans, for each pair of genotype levels. Tukey's HSD also controls the family-wise error rate. We performed Welch two-sample *t*-test's, using R's 't.test' function, to quantify differences in P1NP levels between mutant and wild-type mice.

## Analyses involving data from the International Mouse Phenotyping Consortium

For the IMPC data, we obtained data using their 'statistical-result' SOLR database, using the 'solrium' R package (version 1.1.4) (*Chamberlain, 2019*). We obtained experimental results using the 'Bone*Mineral*Density' parameter. We then pruned the resulting data to only include 'Successful' analyses, and removed experiments that included the skull. To generate the *Gpatch1* boxplot, we obtained raw data using from IMPC's 'statistical-raw-data' SOLR database for *Gpatch1*, and analyzed the data in the same manner as IMPC, using the 'OpenStats' R package (version 1.0.2), using the method = 'MM' and MM_BodyWeightIncluded = TRUE arguments (*Haselimashhadi et al., 2020*). Finally, mouse genes were converted to their human syntenic counterparts using Ensembl's 'hsapiens_gene_ensembl' and 'mmusculus_gene_ensembl' datasets through biomaRt.

## PhenomeXcan data analysis

We obtained all significant PhenomeXcan gene–trait associations from their paper (https://advances.sciencemag.org/content/6/37/eaba2083), and used data for the

'3148_raw-Heel_bone_mineral_density_BMD' phenotype (*Pividori et al., 2020*). Furthermore, we constrained our search to only include genes that were annotated by the authors as 'protein_coding'.

## LSBMD/FNBMD GWAS analysis

We obtained sex-combined LSBMD and FNBMD GWAS summary statistics from GEFOS (http://www.gefos.org/?q=content/data-release-2012), and then used a custom script that utilized the biomaRt R package to convert variants to their GRCh38 coordinates.

## Data availability

eBMD and fracture GWAS summary statistics were obtained from GEFOS, as were the LSBMD and FNBMD GWAS summary statistics. GTEx eQTL data were obtained from the GTEx web portal. Data from the PhenomeXcan project were obtained from *Pividori et al., 2020*. Statistical data from the IMPC were obtained using an R interface to their SOLR database. *Ppp6r3* experimental data are provided on our GitHub (https://github.com/basel-maher/BMD_TWAS_colocalization; *Al-Barghouthi, 2022*). Mouse-Human homologs were obtained from MGI (http://www.informatics.jax.org/downloads/reports/HOM_MouseHumanSequence.rpt). We also obtained data from the MGI Human-Mouse:Disease Connection database (http://www.informatics.jax.org/diseasePortal). Gene Ontologies were obtained from AmiGO2 (http://amigo.geneontology.org/amigo).

## Code availability

Analysis code and the raw data for our *Ppp6r3* functional validation analyses are available on GitHub (https://github.com/basel-maher/BMD_TWAS_colocalization, copy archived at swh:1:rev:6aaa-8819c2e335013a665e76318dc98aeb9a52ce; *Al-Barghouthi, 2022*).

## Acknowledgements

Research reported in this publication was supported in part by the National Institute of Arthritis and Musculoskeletal and Skin Diseases of the National Institutes of Health under Award Number AR071657 to CRF, LCG, and EFM, and by the National Center for Research Resources of the National Institutes of Health under Award Number S10RR021072 to EFM. BMA-B. was supported in part by a National Institutes of Health, Biomedical Data Sciences Training Grant (5T32LM012416). The authors acknowledge Wenhao Xu (University of Virginia) and the Genetically Engineered Mouse Models (GEMM) core for their technical assistance in generating the *Ppp6r3* gene-trap mice. We thank Clifford J Rosen and Phuong T Le (MaineHealth Institute for Research) for performing the P1NP assay. We thank the IMPC for accessibility to BMD data in knockout mice (https://www.mousephenotype.org/). The data used for the analyses described in this manuscript were obtained from the IMPC SOLR database on 3/8/21. The Genotype-Tissue Expression (GTEx) Project was supported by the Common Fund of the Office of the Director of the National Institutes of Health, and by NCI, NHGRI, NHLBI, NIDA, NIMH, and NINDS. The data used for the analyses described in this manuscript were obtained from the GTEx Portal on 6/30/20.

## Additional information

### Competing interests

Cheryl Ackert-Bicknell: Reviewing editor, *eLife*. The other authors declare that no competing interests exist.

### Funding

| Funder | Grant reference number | Author |
| --- | --- | --- |
| National Institute of Arthritis and Musculoskeletal and Skin Diseases | AR071657 | Louis Gerstenfeld |

| Funder | Grant reference number | Author |
|--------|------------------------|--------|
| National Center for Research Resources | S10RR021072 | Jinho Heo Elise Morgan |

The funders had no role in study design, data collection, and interpretation, or the decision to submit the work for publication.

## Author contributions

Basel Maher Al-Barghouthi, Conceptualization, Data curation, Formal analysis, Validation, Investigation, Visualization, Methodology, Writing – original draft, Project administration, Writing – review and editing; Will T Rosenow, Formal analysis; Kang-Ping Du, Resources, Investigation; Jinho Heo, Robert Maynard, Gina Calabrese, Aaron Nakasone, Bhavya Senwar, Investigation; Larry Mesner, Investigation, Writing – review and editing; Louis Gerstenfeld, Virginia Ferguson, Elise Morgan, Supervision, Funding acquisition, Writing – review and editing; James Larner, David Brautigan, Resources, Supervision; Cheryl Ackert-Bicknell, Supervision; Charles R Farber, Conceptualization, Resources, Supervision, Funding acquisition, Methodology, Writing – original draft, Project administration, Writing – review and editing

## Author ORCIDs

Basel Maher Al-Barghouthi (iD) http://orcid.org/0000-0001-9816-8044
Charles R Farber (iD) http://orcid.org/0000-0002-6748-4711

## Ethics

Our animal research protocol was reviewed and approved by the University of Virginia (UVA) Institutional Animal Care and Use Committee (IACUC), approved protocol #3741-12-20, titled 'Generation and characterization of mouse models of osteoporosis'. All experimental work was carried out under our approved UVA Institutional Biosafety Committee protocol #640-08.

## Decision letter and Author response

Decision letter https://doi.org/10.7554/eLife.77285.sa1
Author response https://doi.org/10.7554/eLife.77285.sa2

# Additional files

## Supplementary files

• Supplementary file 1. Supplementary data files associated with this manuscript. (a) 2156 protein-coding genes that are significant by S-MultiXcan transcriptome-wide association study (TWAS) analysis (Bonferroni p value ≤0.05). Columns are Ensembl ID, gene name, the nominal S-MultiXcan p value, the number of tissues available to S-MultiXcan, the number of independent components of variation among the tissues, and the Bonferroni-adjusted p value. (b) 1182 protein-coding genes that are significant in the fastEnloc colocalization analysis (regional colocalization probability [RCP] ≥0.1). Columns are Ensembl ID, signal cluster name from the expression quantitative trait loci (eQTL) analysis, the gene name, the RCP, and the colocalizing Genotype-Tissue Expression (GTEx) tissue. (c) The 512 protein-coding genes that are significant by both TWAS and colocalization (Bonferroni p value ≤0.05 and RCP ≥0.1). Columns are Ensembl ID, signal cluster name from the eQTL analysis, gene name, the RCP, the colocalizing GTEx tissue, the nominal S-MultiXcan p value, and the Bonferroni-adjusted p value. (d) Number of the 512 significantly colocalizing genes per GTEx tissue. Columns are the GTEx tissue and the number of unique colocalizing eQTL in the relevant tissue. (e) The 'known bone gene' list. Columns are gene name and Ensembl ID. (f) The 66 genes that are significant by both TWAS and colocalization (Bonferroni p value ≤0.05 and RCP ≥0.1), and are also members of the 'known bone gene' list. Columns are gene name and Ensembl ID. (g) Gene Ontology (GO) enrichments for the 512 protein-coding genes that are significant by both TWAS and colocalization (Bonferroni p value ≤0.05 and RCP ≥0.1). Only GO enrichments with a p value ≤0.05 are included. Columns are the GO IDs, GO terms, p values, the GO subontology (BP – Biological Process, CC – Cellular Component, MF – Molecular Function), and the Ensembl IDs for the genes that are members of the GO ontology. p values were calculated using a one-sided Fisher's exact test, and were not adjusted for multiple comparisons. (h) The 863 genes closest to estimated bone mineral density (eBMD) genome-wide association study (GWAS) associations. These data were obtained from *Morris et al., 2019*. (i) 137 novel putatively causal BMD genes, after increasing

RCP ≥0.5, and removing genes that were members of the 'known bone gene' list and genes with a nominal (p ≤ 0.05) alteration in BMD as determined by the International Mouse Phenotype Consortium (IMPC). Columns are Gene name, Ensembl ID, colocalization RCP, the tissue with the highest RCP, the S-Multixcan Bonferroni-adjusted p value, and the number of tissues that were significantly colocalizing. Note that TLN2 appears twice due to having the same RCP in two tissues. (j) Results of the differential expression analysis performed on RNA isolated from vertebrae. The analysis results show the difference in expression between mutant and wild-type mice. The columns are the Ensembl IDs, gene names, baseMean (the average of normalized count values, taken over all samples), log2 fold change, standard error of the log2 fold change estimate, the Wald statistic, the test p value, and the BH-adjusted p value.

- Transparent reporting form

### Data availability

All data and source code are available on GitHub: https://github.com/basel-maher/BMD_TWAS_colocalization (copy archived at swh:1:rev:6aaa8819c2e335013a665e76318dc98aeb9a52ce).

The following previously published datasets were used:

| Author(s) | Year | Dataset title | Dataset URL | Database and Identifier |
|---|---|---|---|---|
| Morris JA | 2018 | UK Biobank eBMD and Fracture GWAS Data Release 2018 | http://www.gefos.org/?q=content/data-release-2018 | GEFOS, release-2018 |
| Estrada K | 2012 | Data Release 2012 | http://www.gefos.org/?q=content/data-release-2012 | GEFOS, release-2012 |
| Pividori M | 2020 | PhenomeXcan | http://apps.hakyimlab.org/phenomexcan/ | hakyimlab, phenomexcan |
| GTEx Consortium | 2020 | GTEx V8 | https://gtexportal.org/home/datasets | gtexportal, V8 |
| IMPC Consortium | 2021 | IMPC | https://www.mousephenotype.org/ | mousephenotype, mousephenotype |

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
