## [Editor Report]

Many GWAS studies have been done to understand the genetic contributions to bone density, but very few have managed to pinpoint the gene affected by a polymorphism that caused an observed difference. In this paper, your team shows how scientists can identify causative variants from GWAS studies.

---

## [Decision Letter]

**Decision letter after peer review:**

Thank you for submitting your article "Transcriptome-wide Association Study and eQTL colocalization identify potentially causal genes responsible for bone mineral density GWAS associations" for consideration by *eLife*. Your article has been reviewed by 3 peer reviewers, and the evaluation has been overseen by a Reviewing Editor and Mone Zaidi as the Senior Editor. The following individual involved in the review of your submission has agreed to reveal their identity: Erika Kague (Reviewer #2).

Essential revisions:

The reviewers would like to see more data and discussion on the possible role of LRP5 in the bone phenotype of PPP6R3 mutant mice.

a. Please assess expression levels of Ppp6r3 and Lrp5 in your mutant model.

b. Please include a more in-depth characterization of Ppp6r3 mutant mice:

femoral microCT data, P1NP or TRAP expression.

c. Please discuss whether rs10047483 is in LD with the SNP that is closest to LRP5.

*Reviewer #1 (Recommendations for the authors):*

The authors have presented the methods, data, analysis, and interpretation clearly. It's refreshing that the authors were transparent about several issues that did not show their approach in the most positive light, i.e. the fact that these approaches were not appreciably better than the "nearest neighbor" approach, the fact that deletion of their candidate gene in mouse resulted in the opposite phenotype from the one predicted. In addition, their explanations for these issues were clear and rational. Overall, the manuscript is interesting and informative and, as such, will be useful for researchers in bone biology as well as for many readers interested in the challenge of candidate gene identification from human GWAS.

*Reviewer #2 (Recommendations for the authors):*

I have a few comments for the authors:

"Most loci overlapped one gene (mean = 1.7, median = 1); however, 184 loci overlapped multiple genes, including a locus on Chromosome (Chr.) 20 (lead SNP rs6142137) which contained 9 prioritized genes. (Figure 1D)."

1. Figure 1D does not show the respective locus with 9 prioritized genes. It would be helpful to have Figure1E showing the Associated genomic regions and highlighting the 9 prioritized genes as well as what tissues each of them expressed.

"To do so, we created a database-curated set of genes previously implicated in the regulation of bone processes (henceforth referred to as our "known bone genes" list, N=1,399, Supplementary Table 5)."

2. What database was used to annotate the genes as "known bone genes". Could the authors please clarify? Also, why osteoblast transcriptome was not used here?

"we quantified the number of genes that were both the closest genes to eBMD GWAS associations and were members of the "known bone gene" list. "

3. Please provide additional information on what is considered the closest gene (in Mb).

4. Please provide diagrams showing the different steps of gene prioritization (TWAS+eQTL, "known bone genes" within 1Mb) versus (closest gene + "known bone genes") and their overlap ("Of our 512 prioritized genes, 206 (40%) were also the closest genes to eBMD GWAS associations, with 27 of the remaining 306 prioritized genes (8.8%) being members of the "known bone gene" list").

"Together, these data suggest that many of the genes identified by the combined TWAS/colocalization approach are likely causal BMD GWAS ge "

5. Incomplete phrase.

"An example of one of the 64 genes is GPATCH1, located within a GWAS association on human chromosome 19q13.11. "

6. Please provide a brief description of the gene GPATCH1. Any data showing expression of this gene in bone, other than the phenotype from IMPC?

7. In Figure 3A, could the authors please highlight where PPP6R3 is located?

"Together, these data highlight PPP6R3 as a strong candidate for one of the seven eBMD/fracture associations in this region."

8. Please provide a description of the gene PPP6R3 in the Results section as well.

9. rs10047483 is nearby LRP5, is this SNP in LD with the SNPs closest to LRP5? Do the authors suggest that the association to the LRP5 region is polygenic? In the discussion, could the authors provide further examples of complex BMD loci that could help to accept the also role of PPP6R3 in causality? (ex. WNT16 locus by Chesi et al. doi.org/10.1038/s41467-019-09302-x, another example is the FTO locus)

.

10. Figure 4E: Scales are on the bottom left, not right.

11. Would the lack of information to support Ppp6r3 in fracture risk, would be also due to the age at which the analysis of knockout mice was carried out? Could the authors provide a sentence or two to the discussion?

*Reviewer #3 (Recommendations for the authors):*

While this work makes a solid contribution, the significance is somewhat moderated by the application of established protocols to study eBMD, the limited analysis of 2 genes prioritized using this approach following their systemic ablation, the modest phenotypes of the mutant mice, and the lack of mechanistic studies on candidate gene function. Specifically, one gene was not studied in depth beyond the phenotype available in the IMPC. Deletion of Ppp6r3 causes modest reductions in vertebral BMD measurements by DXA and microCT, but the mechanisms by which the systemic ablation of the Ppp6r3 gene contributes to this mild phenotype are not explored. Additional analysis of Ppp6r3 function in bone cell lines or tissues would be especially important since this 'hit' arose using GTEx, which unfortunately does not include bone tissues and bone cells.

Other points for authors to consider:

The authors acknowledge that the Ppp6r3 variant is near Lrp5, and suggest in the Discussion that the Ppp6r3 variant association is independent of Lrp5. However, given the pivotal and well-known role of Lrp5 in BMD, it is important to document that the effects of the Ppp6r3 gene trap are not affecting Lrp5 expression or function.

Although there is a decrease in femoral width, there is no cortical microCT data for the femur. What might be the cause of this decrease in the width of the femur? Is this a developmental issue? Is there any change in the TRAP or P1NP levels in mutant mice?

It would be great to test whether deletion of PPP6R3 affects the mechanical properties of bone, whether in 3-point bending for the femurs or compression tests for the vertebrae. Does PPP6R3 deletion affect porosity to cause changes in BMD without affecting trabecular parameters?

What is the sex of the mice that were used in this study?

The authors performed pathway analysis for the potentially causal 512 BMD genes and showed GO terms. It would have been informative to include a gene network plot to visualize which genes are shared in these pathways focusing on bone-related pathways or a heatmap plot to show which genes are shared in the bone-related pathways. Additionally, does PPP6R3 belong to any bone-related pathway?

---

## [Author Response]

Essential revisions:The reviewers would like to see more data and discussion on the possible role of LRP5 in the bone phenotype of PPP6R3 mutant mice.a. Please assess expression levels of Ppp6r3 and Lrp5 in your mutant model.

This is a great critique. We performed RNA-seq on RNA isolated from L5 vertebrae from wild-type and *Ppp6r3* mutants. We choose vertebrae for this analysis given the stronger bone phenotype (Figure 4); however, we would expect similar results independent of the bone site analyzed. *Ppp6r3* expression was the most significantly differentially expressed (P=3.67 x 10^-104^, log fold change = -1.63). As would be expected, it was downregulated in *Ppp6r3* mutants (as described in Methods, the mice used in our analysis harbor a gene-trap allele, so while no protein was detected in homozygous mutant mice, transcript would be expected to be present at a low level as observed). We did not observe a difference in *Lrp5* expression or any other gene in proximity (or on the same chromosome) of the gene-trap allele. We have added these data to the manuscript (pg. 8, line 294) and supplemental data file 1j.

b. Please include a more in-depth characterization of Ppp6r3 mutant mice:femoral microCT data, P1NP or TRAP expression.

We agree these additional phenotypes should be addressed. We performed femoral microCT and found significant differences in Total Area and Medullary Area consistent with our caliper-based measures of femoral size. No differences were observed for other cortical or any of the trabecular (distal femur) bone parameters. Additionally, we measured P1NP levels in plasma and observed a significant increase in *Ppp6r3* mutant mice, potentially suggesting high bone turnover in mutant mice. We did not have enough plasma to measure a marker of bone resorption such as TRAP or CTx. We have added these new data to the manuscript (pg. 9, line 336-342), Figure 4 and Figure 4 —figure supplement 2.

c. Please discuss whether rs10047483 is in LD with the SNP that is closest to LRP5.

Of the seven lead eBMD GWAS variants, rs10047483 (lead *PPP6R3* eQTL variant) is only in LD with rs11228240 (r2=0.94) and accounts for the significant colocalization between the *PPP6R3* eQTL and eBMD association. Rs10047483 is NOT in LD (r2<0.11) with any of the additional six eBMD lead GWAS variants, including the ones in close proximity to *LRP5*. These data are stated in the Results section of the manuscript (pg. 7, line 268-277).

Reviewer #2 (Recommendations for the authors):I have a few comments for the authors:"Most loci overlapped one gene (mean = 1.7, median = 1); however, 184 loci overlapped multiple genes, including a locus on Chromosome (Chr.) 20 (lead SNP rs6142137) which contained 9 prioritized genes. (Figure 1D)."1. Figure 1D does not show the respective locus with 9 prioritized genes. It would be helpful to have Figure1E showing the Associated genomic regions and highlighting the 9 prioritized genes as well as what tissues each of them expressed.

We have added Figure 1 —figure supplement 1 which highlights the locus on Chr. 20. Supplementary file 1c contains the tissues, RCPs and TWAS pvals of the genes.

"To do so, we created a database-curated set of genes previously implicated in the regulation of bone processes (henceforth referred to as our "known bone genes" list, N=1,399, Supplementary Table 5)."2. What database was used to annotate the genes as "known bone genes". Could the authors please clarify? Also, why osteoblast transcriptome was not used here?

Good point. We list the databases used in the Methods section entitled, “Creation of the “known bone gene” list”.

We wanted to use genes that had a functional connection to bone, which includes those active in multiple bone cell types and parameters. The bone transcriptome will just show expression in Obs but that doesnt really mean they are “Ob specific”.

"we quantified the number of genes that were both the closest genes to eBMD GWAS associations and were members of the "known bone gene" list. "3. Please provide additional information on what is considered the closest gene (in Mb).

We used the closest gene list published in the Morris et al. BMD GWAS. For each of the 1103 associations identified, the closest gene was defined as the gene whose transcription start site was the shortest distance to the lead variant.

4. Please provide diagrams showing the different steps of gene prioritization (TWAS+eQTL, "known bone genes" within 1Mb) versus (closest gene + "known bone genes") and their overlap ("Of our 512 prioritized genes, 206 (40%) were also the closest genes to eBMD GWAS associations, with 27 of the remaining 306 prioritized genes (8.8%) being members of the "known bone gene" list").

Good idea. We added Figure 2 —figure supplement 1 which includes requested diagrams.

"Together, these data suggest that many of the genes identified by the combined TWAS/colocalization approach are likely causal BMD GWAS ge "5. Incomplete phrase.

Thanks! This has been corrected.

"An example of one of the 64 genes is GPATCH1, located within a GWAS association on human chromosome 19q13.11. "6. Please provide a brief description of the gene GPATCH1. Any data showing expression of this gene in bone, other than the phenotype from IMPC?

We now provide a brief description of GPATCH1 and cite studies demonstrating it is expressed in multiple bone cell-types.

7. In Figure 3A, could the authors please highlight where PPP6R3 is located?

This has been added to Figure 3A.

"Together, these data highlight PPP6R3 as a strong candidate for one of the seven eBMD/fracture associations in this region."8. Please provide a description of the gene PPP6R3 in the Results section as well.

A brief description of PPP6R3 function has been added.

9. rs10047483 is nearby LRP5, is this SNP in LD with the SNPs closest to LRP5? Do the authors suggest that the association to the LRP5 region is polygenic? In the discussion, could the authors provide further examples of complex BMD loci that could help to accept the also role of PPP6R3 in causality? (ex. WNT16 locus by Chesi et al. doi.org/10.1038/s41467-019-09302-x, another example is the FTO locus)

Great points. With regards to the LD, as described above in response to essential revision #3, no, rs10047483 is not in LD with the BMD lead SNPs overlapping LRP5. Yes, we believe the data support a model of alleleic heterogeneity involving multiple genes (at least LRP5 and PPP6R3) within this locus. We have noted the similarity, in terms of complexity, between this locus and the Wnt16 locus in the discussion.

10. Figure 4E: Scales are on the bottom left, not right.

This has been corrected.

11. Would the lack of information to support Ppp6r3 in fracture risk, would be also due to the age at which the analysis of knockout mice was carried out? Could the authors provide a sentence or two to the discussion?

We did not directly measure biomechanical strength of bones from *Ppp6r3* knockout mice; however this could be the focus of future research. We did measure bone matrix composition and, yes, we agree age could be a factor. We have added a brief note in the discussion related to this.

Reviewer #3 (Recommendations for the authors):While this work makes a solid contribution, the significance is somewhat moderated by the application of established protocols to study eBMD, the limited analysis of 2 genes prioritized using this approach following their systemic ablation, the modest phenotypes of the mutant mice, and the lack of mechanistic studies on candidate gene function. Specifically, one gene was not studied in depth beyond the phenotype available in the IMPC. Deletion of Ppp6r3 causes modest reductions in vertebral BMD measurements by DXA and microCT, but the mechanisms by which the systemic ablation of the Ppp6r3 gene contributes to this mild phenotype are not explored. Additional analysis of Ppp6r3 function in bone cell lines or tissues would be especially important since this 'hit' arose using GTEx, which unfortunately does not include bone tissues and bone cells.Other points for authors to consider:The authors acknowledge that the Ppp6r3 variant is near Lrp5, and suggest in the Discussion that the Ppp6r3 variant association is independent of Lrp5. However, given the pivotal and well-known role of Lrp5 in BMD, it is important to document that the effects of the Ppp6r3 gene trap are not affecting Lrp5 expression or function.

Great point. As shown above in the response to essential revision #1, we did not observe a difference in expression of Lrp5 in vertbraes from Ppp6r3 mice.

Although there is a decrease in femoral width, there is no cortical microCT data for the femur. What might be the cause of this decrease in the width of the femur? Is this a developmental issue? Is there any change in the TRAP or P1NP levels in mutant mice?

We have added cortical microCT data of the femur and we did confirm the decrease in bone size. It is unclear if the phenotype is developmental or the mechanism. We also measured P1NP and, surprisingly, we observed a significant increase in mutant mice. In the context of bone mass, this may suggest decreased bone mass due to high turnover. We do want to stress that our goal for this project was using TWAS/colcoalzation to identify candidate genes and then provide additional support for select genes in the form of experimental validation. Now that we have identified Ppp6r3 and confirmed its involvement in bone, we have additional experiments planned to determine how *Ppp6r3* impacts bone and all the points you bring up will be addressed.

It would be great to test whether deletion of PPP6R3 affects the mechanical properties of bone, whether in 3-point bending for the femurs or compression tests for the vertebrae. Does PPP6R3 deletion affect porosity to cause changes in BMD without affecting trabecular parameters?

This is a great point. Unfortunately, we don’t have bones that we could use to measure biomechanical strength and we didn’t measure porosity. Again, you bring up great points that will be critical to address to understand the mechanisms underlying the role of Ppp6r3 in bone that will be addressed in future studies.

What is the sex of the mice that were used in this study?

Thanks for catching this important omission! All studies included both male and female mice. We now specify this in the results, Figure 4 legend, and methods (under *PPP6R3 functional validation*).

The authors performed pathway analysis for the potentially causal 512 BMD genes and showed GO terms. It would have been informative to include a gene network plot to visualize which genes are shared in these pathways focusing on bone-related pathways or a heatmap plot to show which genes are shared in the bone-related pathways. Additionally, does PPP6R3 belong to any bone-related pathway?

We provide all 1140 nominally significant GO enrichments for the 512 genes in Supplementary file 1g, including which of the 512 genes belong to each enriched GO term. Unfortunately, we are not exactly sure what you mean by “network plot”?

No, PPP6R3 does not belong to any explicitly bone related gene ontologies.